# STATISTICAL ADAPTIVE STOCHASTIC OPTIMIZATION

## ABSTRACT

We investigate statistical methods for automatically scheduling the learning rate (step size) in stochastic optimization. First, we consider a broad family of stochastic optimization methods with constant hyperparameters (including the learning rate and various forms of momentum) and derive a general necessary condition for the resulting dynamics to be stationary. Based on this condition, we develop a simple online statistical test to detect (non-)stationarity and use it to automatically drop the learning rate by a constant factor whenever stationarity is detected. Unlike in prior work, our stationarity condition and statistical test apply to different algorithms without modification. Finally, we propose a smoothed stochastic line-search method that can be used to warm up the optimization process before the statistical test can be applied effectively. This removes the expensive trial and error for setting a good initial learning rate. The combined method is highly autonomous and it attains state-of-the-art training and testing performance in our experiments on several deep learning tasks.

## 1 INTRODUCTION

We study adaptive stochastic optimization methods in the context of large-scale machine learning. Specifically, we consider the stochastic optimization problem

$$\underset{x \in \mathbf{R}^p}{\text{minimize}} \quad F(x) \triangleq \mathbf{E}_\xi \big[ f_\xi(x) \big], \tag{1}$$

where $\xi$ is a random variable representing data sampled from some (unknown) probability distribution, $x \in \mathbf{R}^p$ represents the parameters of the machine learning model (e.g., the weight matrices in a neural network), and $f_\xi$ is the loss function associated with data $\xi$. The objective function $F$ is the expectation of $f_\xi$ over the distribution of the data.

Many stochastic optimization methods for solving problem (1) can be written in the form of

$$x^{k+1} = x^k - \alpha_k d^k, \tag{2}$$

where $d^k$ is a (stochastic) search direction and $\alpha_k > 0$ is the step size or learning rate. In the classical stochastic gradient descent (SGD) method,

$$d^k = g^k \triangleq \nabla f_{\xi_k}(x^k), \tag{3}$$

where $\xi_k$ is a training example (or a mini-batch of examples) randomly generated at iteration $k$. This method traces back to the seminal work of Robbins & Monro (1951), and it has become very popular in machine learning (e.g., Bottou, 1998; Goodfellow et al., 2016). Many modifications of SGD aim to improve its theoretical and practical performance. For example, Gupal & Bazhenov (1972) studied a stochastic analog of the heavy-ball method (Polyak, 1964), where

$$d^k = (1 - \beta_k)g^k + \beta_k d^{k-1}, \tag{4}$$

and $\beta_k \in [0, 1)$ is the momentum coefficient. Sutskever et al. (2013) proposed to use a stochastic variant of Nesterov's accelerated gradient method (Nesterov, 2004), where

$$d^k = \nabla f_{\xi_k}(x^k - \alpha_k \beta_k d^{k-1}) + \beta_k d^{k-1}. \tag{5}$$

Other recent variants include, e.g., Jain et al. (2018) and Ma & Yarats (2019).

Theoretical conditions for the asymptotic convergence of SGD are well studied, and they mostly focus on polynomially decaying learning rates of the form $\alpha_k = a/(b + k)^c$ for some $a, b > 0$

and $1/2 < c \leq 1$ (e.g., Robbins & Monro, 1951; Polyak & Juditsky, 1992). Similar conditions for the stochastic heavy-ball methods are also established (e.g., Gupal & Bazhenov, 1972; Polyak, 1977). However, these learning rate schedules still require significant hyperparameter tuning efforts in modern machine learning practice.

Adaptive rules for adjusting the learning rate and other parameters on the fly have been developed in both the stochastic optimization literature (e.g., Kesten, 1958; Mirzoakhmedov & Uryasev, 1983; Ruszczyński & Syski, 1983; 1984; 1986a;b; Delyon & Juditsky, 1993) and by the machine learning community (e.g., Jacobs, 1988; Sutton, 1992; Schraudolph, 1999; Mahmood et al., 2012; Baydin et al., 2018). More recently, adaptive algorithms that use diagonal scaling—replacing $\alpha_k$ in (2) with an adjustable diagonal matrix—have become very popular (Duchi et al., 2011; Tieleman & Hinton, 2012; Kingma & Ba, 2014, e.g.,). Despite these advances, costly hand-tuning efforts are still needed to obtain good (generalization) performance in practice (Wilson et al., 2017).

In this paper, we focus on statistical methods for automatically scheduling the learning rate. This line of work goes back to Kesten (1958), who used the change of sign of the inner products of consecutive stochastic gradients as a statistical indicator of slow progress and as a trigger to decrease the learning rate (see extensions in Delyon & Juditsky, 1993). Pflug (1983; 1988) considered the dynamics of SGD with constant step size for minimizing convex quadratic functions and derived necessary conditions for the resulting process to be stationary. Pflug (1983; 1990) also devised a sequential statistical test to detect stationarity; when the test fires, the step size is decreased by a constant factor. Ruszczyński & Syski (1983) used online statistical tests to check if the present step size and momentum constants satisfy certain optimality conditions and adjust them if the conditions are not satisfied. However, Lang et al. (2019) found that these methods have limited success in machine learning applications due to their reliance on quadratic approximation and/or strong assumptions on the noise models.

Most recently, Yaida (2018) derived fluctuation-dissipation relations (necessary conditions) that characterized the stationary behavior of stochastic gradient methods with constant learning rate and momentum. These relations hold *exactly* for any stationary state (regardless of the loss function or the noise model), so they can be very effective at detecting stationarity and can serve as a reliable indicator of when to decrease the learning rate. Lang et al. (2019) devised a more rigorous statistical test for Yaida's conditions and obtained robust and competitive performance on common deep learning tasks.

Our contributions in this paper are threefold:

- We consider a broad family of stochastic optimization methods with constant hyperparameters (including the learning rate and various forms of momentum) and derive a more general necessary condition (than Yaida's) for the associated learning process to be stationary. Yaida (2018) established a "master equation," from which different stationarity conditions can be derived for different variants of stochastic gradient methods (some of them may not admit a usable analytical form). We derive a simple "master condition" that works for different methods without any change and holds *exactly* for any stationary process. Because of its conceptual and analytical simplicity, it greatly simplifies implementation and deployment in software packages.
- We develop a simple statistical test for checking stationarity based on our "master condition." It is a simple confidence interval test (checking if it contains zero), as opposed to the "equivalence test" for Yaida's condition used by Lang et al. (2019). This simple test is as robust as the equivalence test and avoids the use of an additional hyperparameter.
- A major challenge for almost all adaptive stochastic gradient methods is how to set the initial learning rate appropriately without expensive trial and error. We propose a *smoothed* stochastic line-search method that can be used to warm up the optimization process. Starting from any safe, small learning rate, this method automatically increases it to a suitably large value for fast initial convergence. Afterwards, the statistical test for stationarity can be applied to gradually reduce the learning rate and obtain finer convergence.

Combining the three components above, we obtain a highly autonomous algorithm called SALSA (Stochastic Approximation with Line-search and Statistical Adaptation). We conduct extensive experiments on several deep learning tasks to test its training and testing perfromance as well as its robustness to changes in the hyperparameter settings. SALSA matches the best performance of hand-tuned methods across different deep learning tasks using default hyperparameters.

## 2 NECESSARY CONDITIONS FOR STATIONARITY

We consider general stochastic optimization methods in the form of (2) with a constant learning rate:
$$x^{k+1} = x^k - \alpha d^k. \tag{6}$$
We assume that the stochastic search direction $d^k$ is generated with time-homogeneous dynamics; In particular, any additional hyperparameters involved must be constant (not depending on $k$). As an example, we consider the search directions generated by the family of Quasi-Hyperbolic Momentum (QHM) methods (Ma & Yarats, 2019):
$$h^k = (1 - \beta)g^k + \beta h^{k-1},$$
$$d^k = (1 - \nu)g^k + \nu h^k, \tag{7}$$
where $0 \le \beta < 1$ and $0 \le \nu \le 1$. With $\beta = 0$ or $\nu = 0$, it recovers the SGD direction (3). With $\nu = 1$, it recovers the stochastic heavy-ball direction (4). With $0 < \beta = \nu < 1$, it is equivalent to the direction with Nesterov momentum given in (5) (Sutskever et al., 2013; Gitman et al., 2019). We assume that the dynamics of (6), e.g., driven by the stochastic gradients $g^k$ through (7), is stable. Stability regions of the hyperparameters in QHM are characterized by Gitman et al. (2019).

In addition, we assume that the stochastic process $\{x^k\}$, as $k \to \infty$, becomes *stationary*. Recall that $\{x^k\}$ is (strongly) stationary if the joint distribution of any subset of the sequence is invariant with respect to simultaneous shifts in the time index (e.g., Dembo, 2013, Section 3.2.3). As a direct consequence, for any deterministic function $\phi : \mathbf{R}^p \to \mathbf{R}$, we have
$$\mathbf{E}_\pi\big[\phi(x^{k+1})\big] = \mathbf{E}_\pi\big[\phi(x^k)\big], \qquad \forall\, k, \tag{8}$$
where $\pi$ denotes the stationary distribution of $\{x^k\}$. If the test function $\phi$ is smooth, then we use Taylor expansion to obtain
$$\phi(x^{k+1}) = \phi(x^k - \alpha d^k) = \phi(x^k) - \alpha\langle \nabla\phi(x^k), d^k\rangle + \tfrac{\alpha^2}{2}\langle \nabla^2\phi(x^k)d^k, d^k\rangle + O(\alpha^3).$$
Taking expectations on both sides of the above equality and applying (8), we have
$$\mathbf{E}_\pi\left[\langle \nabla\phi(x^k), d^k\rangle - \tfrac{\alpha}{2}\langle \nabla^2\phi(x^k)d^k, d^k\rangle\right] = O(\alpha^2). \tag{9}$$
For an arbitrary smooth function $\phi$, it is very hard in practice to characterize or approximate the $O(\alpha^2)$ term on the right-hand side. In addition, computing the Hessian-vector product $\nabla^2\phi(x^k)d^k$ can be very costly. Therefore, we only consider simple quadratic functions for which the $O(\alpha^2)$ term in (9) vanishes. In particular, the choice of $\phi(x) = (1/2)\|x\|^2$ results in
$$\mathbf{E}_\pi\left[\langle x^k, d^k\rangle - \tfrac{\alpha}{2}\|d^k\|^2\right] = 0. \tag{10}$$
This condition holds *exactly* for any stochastic optimization method of the form (6) if it reaches stationarity. Indeed, *weak* stationarity is sufficient since $\phi$ is a quadratic function of the state (e.g., Dembo, 2013, Section 3.2.3). Beyond stationarity, it requires no specific assumption on the loss function or noise model for the stochastic gradients. Moreover, it can be applied to different methods without any change. We call this the *master condition* for stationarity.

Yaida (2018) focused on state perturbation along $g^k \triangleq \nabla f_{\xi_k}(x^k)$ to obtain his "master equation"
$$\mathbf{E}_\pi\big[\phi(x^k - \alpha g^k)\big] = \mathbf{E}_\pi\big[\phi(x^k)\big], \qquad \forall\, k.$$
This equation can be combined with the actual state update equation (6) to obtain more specific stationarity conditions for different algorithms. For example, for the stochastic heavy-ball method with $d^k = (1 - \beta)g^k + \beta d^{k-1}$, the master equation leads to (Yaida, 2018)
$$\mathbf{E}_\pi\left[\langle x^k, g^k\rangle - \tfrac{\alpha}{2}\tfrac{1+\beta}{1-\beta}\|d^k\|^2\right] = 0. \tag{11}$$
It is a simple exercise to show that this is equivalent to our master condition (10). For the QHM update (7), a more complex stationarity condition may also be derived, but it will still be equivalent to (10). In practice, the single, simple master condition (10) is much more preferred.

If $\{x^k\}$ starts with a nonstationary distribution and converges to a stationary state, we have
$$\lim_{k\to\infty} \mathbf{E}\left[\langle x^k, d^k\rangle - \tfrac{\alpha}{2}\|d^k\|^2\right] = 0. \tag{12}$$
In the next section, we devise a simple statistical test to determine if the master condition fails to hold. In this case, the learning process has not stalled, and we can continue with the same step size. If we fail to detect non-stationarity (i.e., the dynamics may be approximately stationary), we would like to reduce the learning rate $\alpha$ to allow for finer convergence.

---

**Algorithm 1:** SASA+: SASA with master condition and simple statistical testing

---

1 **input:** $x^0 \in \mathbf{R}^p$, $\alpha_0 > 0$, $N_{\min} > 0$, $K_{\text{test}} > 0$, $\delta \in (0,1)$, $\theta \in (0,1)$, $\tau \in (0,1)$
2 $\alpha \leftarrow \alpha_0$
3 $k_o \leftarrow 0$
4 **for** $k = 0, ..., T-1$ **do**
5     Randomly sample $\xi_k$ and compute stochastic search direction $d^k$ (e.g., using QHM)
6     $x^{k+1} \leftarrow x^k - \alpha d^k$
7     $\Delta_k \leftarrow \langle x^k, d^k \rangle - \frac{\alpha}{2}\|d^k\|^2$
8     $N \leftarrow \lceil \theta(k - k_o) \rceil$
9     **if** $N > N_{\min}$ **and** $k \mod K_{\text{test}} == 0$ **then**
10        $(\hat{\mu}_N, \hat{\sigma}_N) \leftarrow$ sample mean and BM/OLBM variance of $\{\Delta_{k-N+1}, \ldots, \Delta_k\}$
11        **if** $0 \in \hat{\mu}_N \pm t^*_{1-\delta/2}\frac{\hat{\sigma}_N}{\sqrt{N}}$ **then**
12           $\alpha \leftarrow \tau \alpha$
13           $k_o \leftarrow k$
14        **end**
15     **end**
16 **end**
17 **output:** $x^T$ (or the average of last epoch)

---

Table 1: List of hyperparameters of Algorithm 1

| Parameter | Range | Explanation | Default value |
|---|---|---|---|
| $N_{\min}$ | $\mathbb{Z}_+$ | minimum number of samples for testing | $\min\{1000, \lceil n/b \rceil\}$ |
| $K_{\text{test}}$ | $\mathbb{Z}_+$ | period to perform statistical test | $\min\{100, \lceil n/b \rceil\}$ |
| $\delta$ | $(0,1)$ | $(1-\delta)$-confidence interval | 0.01 (99% confidence) |
| $\theta$ | $(0,1)$ | fraction of recent samples to keep (after reset) | $1/4$ |
| $\tau$ | $(0,1)$ | learning rate drop factor | $1/10$ |

## 3 STATISTICAL TESTS OF (NON-)STATIONARITY

In order to test if the master condition (10) holds approximately, we collect the simple statistics

$$\Delta_k \triangleq \langle x^k, d^k \rangle - \tfrac{\alpha}{2}\|d^k\|^2 \; = \; \langle x^{k+1}, d^k \rangle + \tfrac{\alpha}{2}\|d^k\|^2.$$

Here the second expression for $\Delta_k$ is obtained using the direct substitution $x^{k+1} = x^k - \alpha d^k$, which can be more convenient to implement if $\Delta_k$ is collected after the state updates to $x^{k+1}$.

In the language of statistical hypothesis testing (e.g., Lehmann & Romano, 2005), we make as our *null hypothesis* that the dynamics (6) have reached a stationary distribution $\pi$. If we have $N$ samples $\{\Delta_k\}$, we know from equation (10) and the Markov chain CLT (see its application in Jones et al. (2006)) that as $N \to \infty$, under the null hypothesis the mean statistic $\bar{\Delta}$ follows a normal distribution with mean 0 and variance $\sigma_\Delta^2/\sqrt{N}$. Our *alternative hypothesis* is that the dynamics (6) have *not* reached stationarity. To test these hypotheses, we adopt the classical confidence interval test. More specifically, we use the most recent $N$ samples $\{\Delta_{k-N-1}, \ldots, \Delta_k\}$ to compute the sample mean $\hat{\mu}_N$ and a variance estimator $\hat{\sigma}_N^2$ for $\sigma_\Delta^2$. Then we can form the $(1-\delta)$-confidence interval

$$(\hat{\mu}_N - \omega_N, \; \hat{\mu}_N + \omega_N) \qquad \text{with half width} \qquad \omega_N = t^*_{1-\delta/2}\frac{\hat{\sigma}_N}{\sqrt{N}}, \qquad (13)$$

where $t^*_{1-\delta/2}$ is the $(1 - \delta/2)$ quantile of the Student's $t$-distribution with degrees of freedom corresponding to that in the variance estimator $\hat{\sigma}_N^2$. Because the sequence $\{\Delta_{k-N+1}, \ldots, \Delta_k\}$ are highly correlated due to the underlying Markov dynamics, the classical formula for the sample variance (obtained by assuming i.i.d. samples) does not work for $\hat{\sigma}_N^2$. We need to use more sophisticated *batch mean* (BM) or *overlapping batch mean* (OLBM) variance estimators developed in the Markov chain Monte Carlo literature (e.g., Jones et al., 2006; Flegal & Jones, 2010). See Lang et al. (2019) for detailed explanation. We also list the formulas for computing BM and OLBM in Appendix A.

---

**Algorithm 2:** Smoothed Stochastic Line-Search (SSLS)

---

**1 input:** $x^0 \in \mathbf{R}^p$, $\alpha_{-1} > 0$, sufficient decrease coefficient $c \in (0, 1/2)$, line-search factors
$\quad\quad \rho_{\text{inc}} \geq 1$, $\rho_{\text{dec}} \in (0, 1)$, smoothing parameter $\gamma \in [0, 1]$ and maximum LS count $m$
**2 for** $k = 0, ..., T - 1$ **do**
**3** $\quad$ Sample $\xi_k$, compute $g^k \leftarrow \nabla f_{\xi_k}(x^k)$ and search direction $d^k$ (e.g., using QHM)
**4** $\quad$ $\eta_k \leftarrow \rho_{\text{inc}} \alpha_{k-1}$
**5** $\quad$ **while** $f_{\xi_k}(x^k - \eta_k g^k) > f_{\xi_k}(x^k) - c \cdot \eta_k \|g^k\|^2$ **and** $\eta_k > \rho_{\text{dec}}^m \alpha_{k-1}$ **do**
**6** $\quad\quad$ $\mid$ $\eta_k \leftarrow \rho_{\text{dec}} \eta_k$
**7** $\quad$ **end**
**8** $\quad$ $\alpha_k \leftarrow (1 - \gamma)\alpha_{k-1} + \gamma \eta_k$
**9** $\quad$ $x^{k+1} \leftarrow x^k - \alpha_k d^k$
**10 end**
**11 output:** $x^T$

---

If the confidence interval in (13) contains 0, we fail to reject the null hypothesis that the learning process is at stationarity, which means the learning rate should be decreased. Otherwise, we accept the alternative hypothesis of non-stationarity and keep using the current constant learning rate.

Stochastic optimization methods equipped with such an adaptation scheme belong to the general framework of SASA (Statistical Adaptive Stochastic Approximation) proposed by Lang et al. (2019). Algorithm 1 is our new variant called SASA+. Table 1 lists its hyperparameters and their default values (where $n$ is total number of training examples and $b$ is the mini-batch size).

Lang et al. (2019) focused on the stochastic heavy-ball method and devised a statistical test for the condition (11). Another major difference is that they set non-stationarity as the null hypothesis and stationarity as the alternative hypothesis (opposite to ours). This leads to a more complex "equivalence test" that requires an additional hyperparameter. Our test is simpler and more straightforward, and computationally as robust.

## 4 SMOOTHED STOCHASTIC LINE SEARCH

SASA+ can automatically decrease the learning rate to refine the last phase of the optimization process, but it relies on an appropriate setting of the starting learning rate $\alpha_0$ for sufficient initial exploration. The appropriate initial learning rate varies substantially for different objective functions, machine learning models, and training datasets, and setting it correctly without expensive trial and error is a major challenge for all stochastic gradient methods, adaptive or not.

Several recent works (e.g., Schmidt et al., 2017; Vaswani et al., 2019) explore the use of classical line-search techniques (e.g., Nocedal & Wright, 2006, Chapter 3) for solving stochastic optimization problems with special structure. One of the main difficulties of applying line-search to stochastic optimization is that the estimated step sizes may vary a lot from step to step and it may not capture the appropriate step size for the average loss function. To overcome this difficulty, we propose a smoothed stochastic line-search (SSLS) procedure listed in Algorithm 2.

During each step $k$, SSLS uses the classical Armijo line-search to find an appropriate step size $\eta_k$ for the randomly chosen function $f_{\xi_k}$ (initialized by $\alpha_{k-1}$), then sets the overall learning rate to be

$$\alpha_k = (1 - \gamma)\alpha_{k-1} + \gamma \eta_k,$$

where $\gamma \in [0, 1]$. When $\gamma = 1$, SSLS reduces to the stochastic Armijo line-search used by Vaswani et al. (2019). A good choice is to set $\gamma = b/n$ where $n$ is the total number of training examples and $b$ is the mini-batch size. Suppose $\rho_{\text{inc}} = 2$ and $\eta_k = 2\alpha_{k-1}$ is accepted at step $k$, then

$$\alpha_k = (1 - \gamma)\alpha_{k-1} + \gamma(2\alpha_{k-1}) = (1 + \gamma)\alpha_{k-1}.$$

If this happens at every iteration over one epoch (of $\lceil n/b \rceil$ iterations) and $n \gg b$, then

$$\alpha_{k+\lceil n/b \rceil} = (1 + b/n)^{\lceil n/b \rceil} \alpha_k \approx e \cdot \alpha_k. \tag{14}$$

Therefore, the most aggressive growth of the learning rate is by a factor of $e$ over one epoch. Such a growing factor is reasonable for line search in deterministic optimization (e.g., Nesterov, 2013).

---

**Algorithm 3:** SALSA: SASA+ with warmup by SSLS

---

**1 input:** $x^0 \in \mathbf{R}^p$, number of steps for warmup $T_{\text{wu}}$ and total number of steps $T > T_{\text{wu}}$
**2 for** $k = 0, ..., T_{\text{wu}}$ **do**
**3** | Run SSLS (Algorithm 2)
**4 end**
**5 for** $k = T_{\text{wu}} + 1, ..., T$ **do**
**6** | Run SASA+ (Algorithm 1)
**7 end**
**8 output:** $x^T$

---

Setting $\gamma = \sqrt{b/n}$ leads to maximum growth of $e^2$ over one epoch. A similar smoothing effect holds for decreasing the learning rate as well. The smoothing scheme allows us to use standard increasing and decreasing factors, such as $\rho_{\text{inc}} = 2$ and $\rho_{\text{dec}} = 1/2$. Without the smoothing scheme, Vaswani et al. (2019) set $\rho_{\text{inc}} = 2^{b/n}$ to restrict dramatic growth of $\alpha_k = \eta_k$ (equivalent to $\gamma = 1$ in SSLS). However, the decreasing of $\alpha_k = \eta_k$ can be excessive and premature, even with $\rho_{\text{dec}} = 0.9$.

Following the theoretical framework of Vaswani et al. (2019), it may be possible to show that SSLS has similar convergence properties as stochastic Armijo line-search under the smooth and interpolation assumptions, which we leave as a future research project. In this paper, we are mainly interested in investigating its performance in practice. In particular, we use it on deep neural network models with ReLU activations. Here the loss functions are non-smooth, and classical theoretical foundations for line-search do not carry over. Nevertheless, we found SSLS to have robust performance in all of our experiments. In order to handle the case of potential non-descent directions in the non-smooth case, we exit the line search after a maximum of $m$ tries by adding the condition $\eta_k > \rho_{\text{dec}}^m \eta_k$ for staying in the line-search loop. By default, we set $m = 10$ together with $\rho_{\text{dec}} = 1/2$.

Finally, we combine SASA+ with SSLS to form Algorithm 3, which we call SALSA (Stochastic Approximation with Line-search and Statistical Adaption). Without prior knowledge on the loss function and training dataset, we start with a very small learning rate and use SSLS to gradually increase it to be around a stationary value that is (automatically) customized to the problem. The after $T_{\text{wu}}$ steps, we switch to SASA+ to gradually reduce the learning rate to settle down to a (hopefully good) local minimum. We typically set the number of warm-up steps $T_{\text{wu}}$ to be equivalent to a few tens of epochs. According to the calculation in (14), using 30 epochs would allow a potential growth factor up to a reasonable fraction of $e^{30}$, which is sufficient for most applications. We could avoid a warmup phase by running SASA+ and SLSS together until the first SASA+ drop, which could only come after SLSS had reached a relatively stable value for $\alpha_k$, allowing the dynamics to approach stationarity. However, we use a fixed warmup phase $T_{\text{wu}}$ in our experiments for simplicity.

**Computational cost of SSLS.** When we fix the number of training epochs, the wall-clock time of Smoothed Stochastic Line Search (SSLS) is about 1.5 times of that of the plain stochastic optimization algorithm. This 0.5 computational overhead is due to 2 function evaluations in each line search step *on average*. Notice that the function evaluation is on the same minibatch and line search only needs the function value without gradient. In SALSA, we switch from SSLS to SALSA at one-third of the total training epochs. Therefore, the total computational overhead caused by using SSLS is only 0.15 of the original total training time.

## 5 EXPERIMENTS

We evaluate the performance of SASA+ (Algorithm 1), SSLS (Algorithm 2) and SALSA (Algorithm 3) by conducting several experiments on two common deep learning datasets. We compare SALSA to the following baselines: SGD with a hand-tuned constant-and-cut learning rate schedule and Adam with a tuned warmup phase (e.g., Wilson et al., 2017; Lang et al., 2019). We also compare SASA+ to the original SASA from Lang et al. (2019). We use weight decay in all experiments.

We use the default values in Table 1 for hyperparameters.

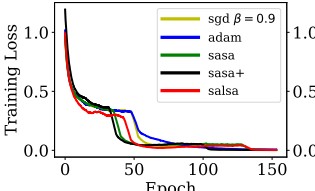 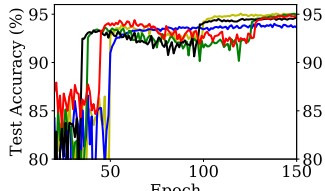 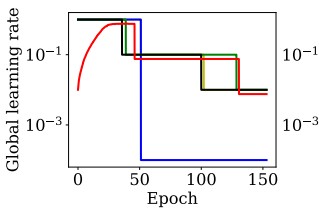

Figure 1: Comparison of different optimizers on CIFAR-10. All SASA, SASA+ and SALSA are using SGD with momentum 0.9. SALSA starts from a small initial learning rate 0.01 while other methods starts from an oracle maximal learning rate 1.0. SALSA witches from SSLS to SASA+ at epoch 40.

**CIFAR-10** We trained an 18-layer ResNet model (He et al., 2016) on CIFAR-10 (Krizhevsky & Hinton, 2009) with random cropping and random horizontal flipping for data augmentation and weight decay 0.0005. Row 1 of Figure 1 compares the best performance of each method. Here SGM-hand uses $\alpha_0 = 1.0$ and $\beta = 0.9$ and drops $\alpha$ by a factor of 10 every 50 epochs. Adam has a tuned global learning rate $\alpha_0$ and a "warmup" phase of 50 epochs. With only a tuned $\alpha_0 = 1e^{-4}$ (the optimal value in our grid search of $\{1e^{-5}, 1e^{-4}, 1e^{-3}, 1e^{-2}\}$), Adam is only able to reach around 93% test accuracy for this model. On the other hand, both SASA+ and SALSA are able to reach similar performance with the hand-tuned SGD. While other methods starts from an oracle maximal learning rate 1.0, SALSA starts from a small initial learning rate 0.01. The SSLS in the first stage of SALSA is able to gradually grow the learning rate to a maximal value and then switches to SASA+ in the second stage.

Figure 2 shows the evolution of SASA+'s different statistics over the course of training the ResNet18 model on CIFAR-10 using the default parameters in Table 1. Between two jumps, the statistics $\Delta_k$ decays toward zero. As long as its confidence interval (13) does *not* overlap with 0, we are confident that the process is not stationary yet and keep exploring with the current learning rate. Otherwise, we decrease the learning rate and enter another cycle of approaching stationarity.

We give experiments that demonstrate the sensitivity of SASA+ to values around its defaults in Figure 3. The top row shows that SASA+ is robust to the choice of the dropping factor $\tau$. The larger the $\tau$ is, the longer the process will stay between two drops, which can be also seen in Figure 2. The middle row shows the effect of statistical parameter $\delta$. Smaller $\delta$ leads to wider confidence interval, and makes easier and thus earlier to fire the statistical test. The bottom row shows the effect of $\theta$, the fractions of samples to keep after reset. Smaller $\theta$ values lead to more frequent dropping, but do not seem to impact the final performance.

Figure 4 shows the dynamics of SSLS and SALSA. In the upper row, SLSS increases the learning rate in the initial phase, and then reaches a stable learning rate once the increases and decrease balance out. With enough momentum (QHM $\beta = \nu = 0.9$), the SSLS can further decrease the learning rate automatically and achieve good training and testing performance. However, without momentum (SGD), the stable learning rate gets too large, and its optimization performance is bad, but the process is still stable and the learning rate does not blow up. In the lower row, SALSA switches from SSLS to SASA+ when it reaches the maximal learning rate. Then even in the case without momentum, SASA+ still works and finally gets as good performance as best hand-tuning SGD. This shows that the SSLS is able to approximate the best hand-tuned maximal learning rate, and the combination of SSLS and SASA+ is necessary to get final good optimization performance.

**ImageNet** We also test our algorithms on the large scale ImageNet dataset (Deng et al., 2009). We use the ResNet18 architecture, random cropping and random horizontal flipping for data augmentation and weight decay 0.0001. Row 1 of Figure 5 compares the performance of the different optimizers. Even with a tuned $\alpha_0 = 1e^{-4}$ (the optimal value in our grid search of $\{1e^{-5}, 1e^{-4}, 1e^{-3}, 1e^{-2}\}$) and allowed to have a long warmup phase (30 epochs), Adam failed to match the generalization performance of SGM. On the other hand, both SASA+ and SALSA were able to match the performance of hand-tuned SGM using the default values of its parameters. Although both SASA+ and SALSA are using the NAG (equivalent to QHM (7) with $\beta = \nu = 0.9$),

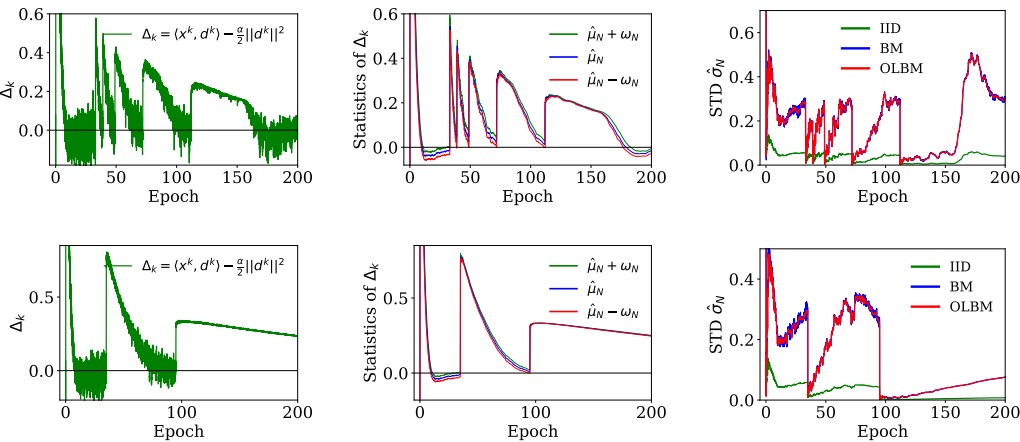

Figure 2: Statistics used in SASA+. Upper row: drop ratio $\tau = 1/2$; lower row: $\tau = 1/10$. The left column shows the instantaneous value of $\Delta_k$. The middle column shows the confidence interval of $\mathbf{E}[\Delta_k]$ constructed by SASA+, which should contain 0 with high probability if the process is stationary. The right column shows the variance estimated by different methods, where BM and OLBM takes into the consideration of correlation in Markov chains and are more accurate.

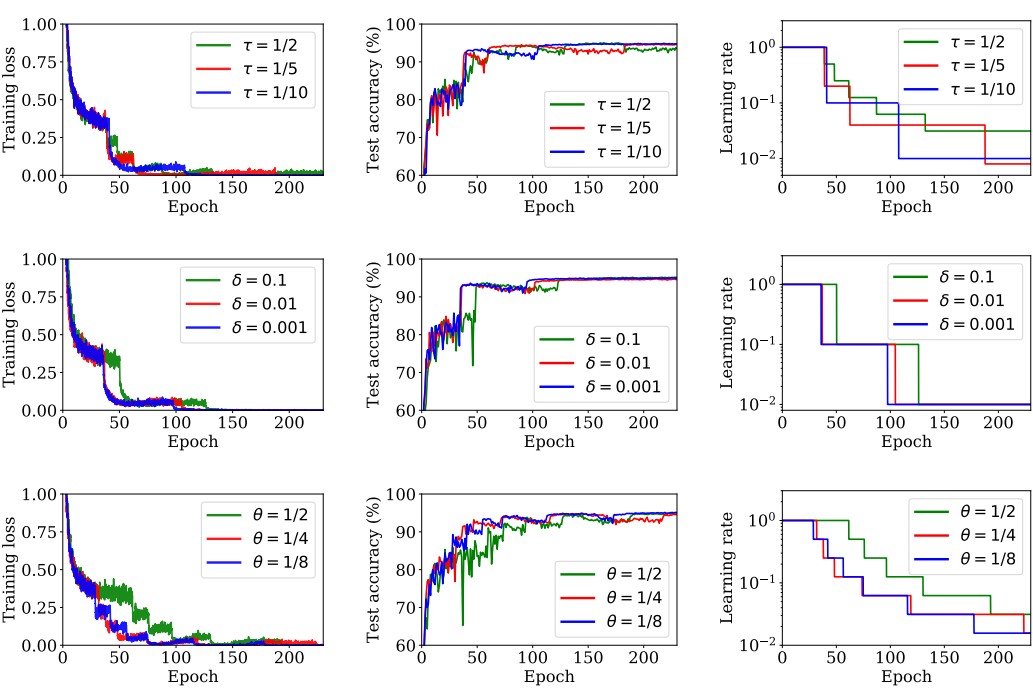

Figure 3: Sensitivity analysis of SASA+ on CIFAR10, using $\beta = 0.9$ and $\nu = 1$. The training loss, test accuracy, and learning rate schedule for SASA+ using different values of $\tau$ (top row), $\delta$ (middle row), and $\theta$ (bottom row) around the default values are shown.

they start with different initial learning rates and have quite different dynamics. SASA+(Nag) starts with an initial learning rate of 1.0, while the SALSA(Nag) starts with a small learning rate 0.1 and automatically determines the appropriate largest learning rate by using SSLS. As a consequence, SASA+(Nag) reaches a stationary state much faster. Nevertheless, their final performances are nearly the same, with SALSA(Nag) even slightly higher.

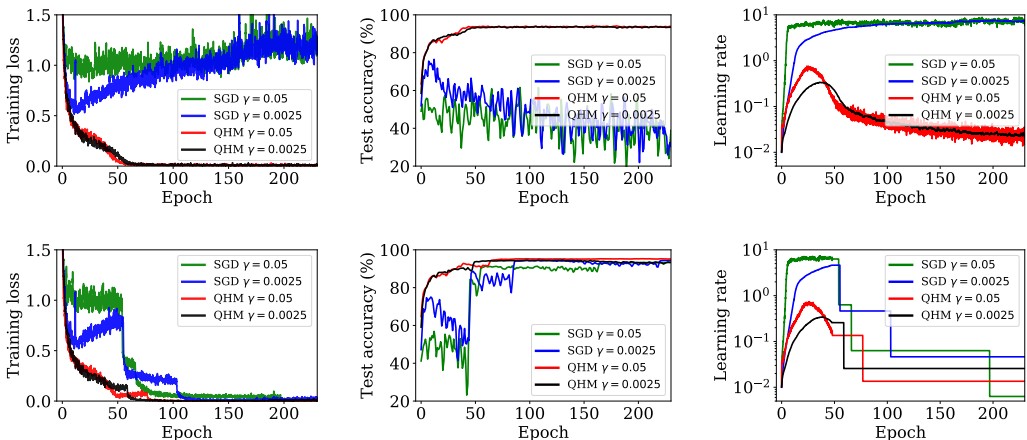

Figure 4: First row: SSLS. Second row: SALSA (warmup 40 epochs). Here, SGD with $\beta = 0.0$ and QHM with parameters $\beta = 0.9$ and $\nu = 0.9$. SSLS is able to find a balance between increasing and decreasing learning rate, and results in a stable large learning rate (SGD) or further decreases the learning rate (QHM). In either case, switching to SASA+ (the lower row) gets good final performance.

Row 2 of Figure 5 shows that SASA+ is simple and general enough to work with different stochastic optimization methods with constant hyperparameters, i.e., SGM with $\beta = 0.9$, NAG, and the recommended QHM setting ($\beta = 0.999$ and $\nu = 0.7$). The dotted curves show the performance of these methods with the best hand-tuned constant-and-cut learning rate schedule. The solid lines show the automatic tuning by SASA+, which learns faster and finally achieves similar performance.

Figure 6 shows the sensitivity analysis of SSLS. From the left and middle panels, we can see that the SSLS always converges to a large learning rate (1.806 in this case), which is comparable with the best human-chosen value of 1.0. The right panel shows that the larger the sufficient descent constant $c$ is, the smaller the maximal learning rate is and the slower SSLS reaches to its maximal learning rate. This is intuitive because larger $c$ puts more limits on the SSLS to grow the learning rate. Moreover, within 60 epochs, SSLS never decreases the learning rate again once it achieves a large stable learning rate, so it is necessary to switch to SASA+ after SLSS reaches a stable learning rate.

## 6 CONCLUSIONS

We presented SASA+, a simpler yet more powerful variant of the statistical adaptive stochastic approximation (SASA) method proposed by Lang et al. (2019). SASA+ is equipped with a single statistical test for (non-)stationarity that works for a broad family of stochastic optimization methods without modification. This greatly simplifies its implementation and deployment in software packages. While SASA+ focuses on how to automatically reducing the learning rate to obtain better asymptotic convergence, we also propose a smoothed stochastic line-search method (SSLS) to warm up the optimization process, thus removing the burden of expensive trial and error for setting a good initial learning rate. The combined algorithm, SALSA, is highly autonomous and robust over variations of its hyperparameters. Using the same default setting, SALSA obtained state-of-the-art performance on several common deep learning models that is competitive with the best hand-tuned optimizers.

## REFERENCES

Atilim Günes Baydin, Robert Cornish, David Martínez Rubio, Mark Schmidt, and Frank Wood. Online learning rate adaptation with hypergradient descent. In *Proceedings of the Sixth International Conference on Learning Representations (ICLR)*, Vancouver, Canada, 2018.

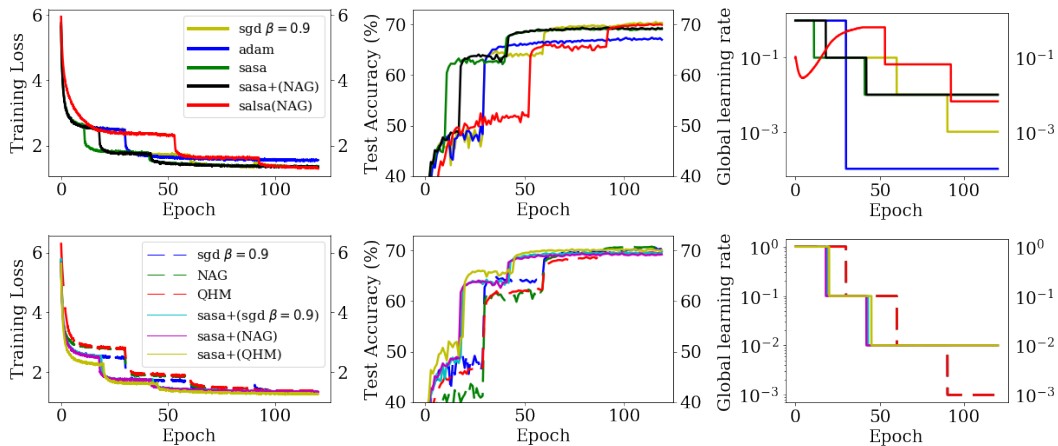

Figure 5: Upper: Comparison of different optimizers on ImageNet. All SASA runs use SGD with momentum 0.9, while SASA+ and SALSA use NAG (QHM with $\beta = \nu = 0.9$). SALSA starts from a small initial learning rate 0.1 while other methods starts from an oracle maximal learning rate 1.0. SALSA witches from SSLS to SASA+ at epoch 40. Lower: the first three curves are stochastic optimization algorithms with hand-tuning learning rate, i.e., decrease every 30 epochs as it is shown in the lower right panel. The last three curves are SASA+ combined with these 3 algorithms. SASA+ automatically adapt their learning rate (see the lower right panel), achieves comparable and even slightly higher testing accuracy (see the lower middle panel).

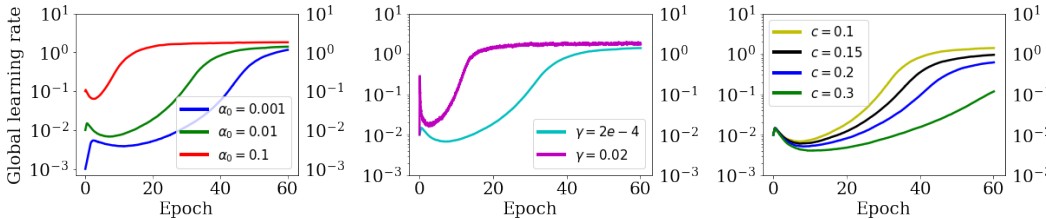

Figure 6: Sensitivity analysis of SSLS around $\alpha_0 = 0.01, \gamma = b/n = 0.0002, c = 0.1$, with SGD $\beta = 0.9, \nu = 1$ on ImageNet.

Léon Bottou. Online algorithms and stochastic approximations. In David Saad (ed.), *Online Learning and Neural Networks*. Cambridge University Press, Cambridge, UK, 1998. URL http://leon.bottou.org/papers/bottou-98x. Revised, Oct 2012.

B. Delyon and A. Juditsky. Accelerated stochastic approximation. *SIAM Journal on Optimization*, 3(4):868–881, 1993.

Amir Dembo. Stochastic Processes. Lecture notes, Stanford University, 2013.

J. Deng, W. Dong, R. Socher, L.-J. Li, K. Li, and L. Fei-Fei. ImageNet: A Large-Scale Hierarchical Image Database. In *CVPR09*, 2009.

John Duchi, Elad Hazan, and Yoram Singer. Adaptive subgradient methods for online learning and stochastic optimization. *Journal of Machine Learning Research*, 12(Jul):2121–2159, 2011.

James M Flegal and Galin L Jones. Batch means and spectral variance estimators in markov chain monte carlo. *The Annals of Statistics*, 38(2):1034–1070, 2010.

Igor Gitman, Hunter Lang, Pengchuan Zhang, and Lin Xiao. Understanding the role momentum in stochastic gradient methods. In *Advances in Neural Information Processing Systems*, volume 32, 2019.

Ian Goodfellow, Yoshua Bengio, and Aaron Courville. *Deep Learning*. MIT Press, 2016. `http://www.deeplearningbook.org`.

A. M. Gupal and L. T. Bazhenov. A stochastic analog of the conjugate gradient method. *Cybernetics*, 8(1):138–140, 1972.

Kaiming He, Xiangyu Zhang, Shaoqing Ren, and Jian Sun. Deep residual networks for image recognition. In *Proceedgins of the 29th IEEE Conference on Computer Vision and Pattern Recognition (CVPR)*, pp. 770–778, 2016.

R. A. Jacobs. Increased rates of convergence through learning rate adaption. *Neural Networks*, 1: 295–307, 1988.

Prateek Jain, Sham M Kakade, Rahul Kidambi, Praneeth Netrapalli, and Aaron Sidford. Accelerating stochastic gradient descent for least squares regression. In *Conference On Learning Theory*, pp. 545–604, 2018.

Galin L Jones, Murali Haran, Brian S Caffo, and Ronald Neath. Fixed-width output analysis for markov chain monte carlo. *Journal of the American Statistical Association*, 101(476):1537–1547, 2006.

Harry Kesten. Accelerated stochastic approximation. *Annals of Mathematical Statistics*, 29(1): 41–59, 1958.

Diederik P Kingma and Jimmy Ba. Adam: A method for stochastic optimization. *arXiv preprint arXiv:1412.6980*, 2014.

Alex Krizhevsky and Geoffrey Hinton. Learning multiple layers of features from tiny images. Technical report, Citeseer, 2009.

Hunter Lang, Pengchuan Zhang, and Lin Xiao. Using statistics to automate stochastic optimization. In *Advances in Neural Information Processing Systems*, volume 32, 2019.

Erich L. Lehmann and Joseph P. Romano. *Testing Statistical Hypotheses*. Springer, 3rd edition, 2005.

Jerry Ma and Denis Yarats. Quasi-hyperbolic momentum and adam for deep learning. In *International Conference on Learning Representations*, 2019.

Ashique Rupam Mahmood, Richard S. Sutton, Thomas Degris, and Patrick M. Pilarski. Tuning-free step-size adaption. In *Proceedings of the IEEE International Conference on Acoustics, Speech and Signal Processing (ICASSP)*, pp. 2121–2124, 2012.

F. Mirzoakhmedov and S. P. Uryasev. Adaptive step adjustment for a stochastic optimization algorithm. *Zh. Vychisl. Mat. Mat. Fiz.*, 23(6):1314–1325, 1983. [U.S.S.R. Comput. Math. Math. Phys. 23:6, 1983].

Y. Nesterov. Gradient methods for minimizing composite objective function. *Mathematical Programming, Series B*, 140:125–161, 2013.

Yurii Nesterov. *Introductory Lecture on Convex Optimization: A Basic Course*. Kluwer Academic Publishers, 2004.

Jorge Nocedal and Stephen J. Wright. *Numerical Optimization*. Springer, 2nd edition, 2006.

Georg Ch. Pflug. On the determination of the step size in stochastic quasigradient methods. Collaborative Paper CP-83-025, International Institute for Applied Systems Analysis (IIASA), Laxenburg, Austria, 1983.

Georg Ch. Pflug. Adaptive stepsize control in stochastic approximation algorithms. In *Proceedings of 8th IFAC Symposium on Identification and System Parameter Estimation*, pp. 787–792, Beijing, 1988.

Georg Ch. Pflug. Non-asymptotic confidence bounds for stochastic approximation algorithms with constant step size. *Monatshefte für Mathematik*, 110:297–314, 1990.

Boris T. Polyak. Some methods of speeding up the convergence of iteration methods. *USSR Computational Mathematics and Mathematical Physics*, 4(5):1–17, 1964.

Boris T. Polyak. Comparison of the rates of convergence of one-step and multi-step optimization algorithms in the presence of noise. *Engineering Cybernetics*, 15:6–10, 1977.

Boris T Polyak and Anatoli B Juditsky. Acceleration of stochastic approximation by averaging. *SIAM Journal on Control and Optimization*, 30(4):838–855, 1992.

Herbert Robbins and Sutton Monro. A stochastic approximation method. *The Annals of Mathematical Statistics*, 22(3):400–407, 1951.

Andrzej Ruszczyński and Wojciech Syski. Stochastic approximation method with gradient averaging for unconstrained problems. *IEEE Transactions on Automatic Control*, 28(12):1097–1105, 1983.

Andrzej Ruszczyński and Wojciech Syski. Stochastic approximation algorithm with gradient averaging and on-line stepsize rules. In J. Gertler and L. Keviczky (eds.), *Proceedings of 9th IFAC World Congress*, pp. 1023–1027, Budapest, Hungary, 1984.

Andrzej Ruszczyński and Wojciech Syski. On convergence of the stochastic subgradient method with on-line stepsize rules. *Journal of Mathematical Analysis and Applications*, 114:512–527, 1986a.

Andrzej Ruszczyński and Wojciech Syski. A method of aggregate stochastic subgradients with on-line stepsize rules for convex stochastic programming problems. *Mathematical Programming Study*, 28:113–131, 1986b.

Mark Schmidt, Nicolas Le Roux, and Francis Bach. Minimizing finite sums with the stochastic average gradient. *Mathematical Programming*, 162(1-2):83–112, 2017.

Nicol N. Schraudolph. Local gain adaptation in stochastic gradient descent. In *Proceedings of Nineth International Conference on Artificial Neural Networks (ICANN)*, pp. 569–574, 1999.

David L Streiner. Unicorns do exist: A tutorial on proving the null hypothesis. *The Canadian Journal of Psychiatry*, 48(11):756–761, 2003.

Ilya Sutskever, James Martens, George Dahl, and Geoffrey Hinton. On the importance of initialization and momentum in deep learning. In Sanjoy Dasgupta and David McAllester (eds.), *Proceedings of the 30th International Conference on Machine Learning*, volume 28 of *Proceedings of Machine Learning Research*, pp. 1139–1147, Atlanta, Georgia, USA, 17–19 Jun 2013. PMLR.

Richard S. Sutton. Adapting bias by gradient descent: An incremental version of Delta-Bar-Delta. In *Proceedings of the Tenth National Conference on Artificial Intelligence (AAAI'92)*, pp. 171–176. The MIT Press, 1992.

Tijmen Tieleman and Geoffrey Hinton. Lecture 6.5-rmsprop: Divide the gradient by a running average of its recent magnitude. *COURSERA: Neural networks for machine learning*, 4(2):26–31, 2012.

Sharan Vaswani, Aaron Mishkin, Issam Laradji, Mark Schmidt, Gauthier Gidel, and Simon Lacoste-Julian. Painless stochastic gradient: Interpolation, line-search, and convergence rates. In *Advances in Neural Information Processing Systems*, volume 32, 2019.

Ashia C Wilson, Rebecca Roelofs, Mitchell Stern, Nati Srebro, and Benjamin Recht. The marginal value of adaptive gradient methods in machine learning. In *Advances in Neural Information Processing Systems*, pp. 4148–4158, 2017.

Sho Yaida. Fluctuation-dissipation relations for stochastic gradient descent. *arXiv preprint arXiv:1810.00004*, 2018.

## A  MCMC VARIANCE ESTIMATORS

Several estimators for the asymptotic variance of the history-average estimator of a Markov chain have appeared in work on Markov Chain Monte Carlo (MCMC). Jones et al. (2006) gives a nice example of such results. Here we simply list two common variance estimators, and we refer the reader to that work and Lang et al. (2019) (from which we borrow notation) for appropriate context and formality.

**Batch means variance estimator.**  Let $\bar{z}_N$ be the history average estimator with $N$ samples, that is, given samples $\{z_i\}$ from a Markov chain, $\bar{z}_N = \frac{1}{N}\sum_{i=i_0}^{i_0+N} z_i$. Now form $p$ batches from the $N$ samples, each of size $q$. Compute the "batch means" $\bar{z}^j = \frac{1}{q}\sum_{i=jq}^{(j+1)q-1} z_i$ for each batch $j$. Then compute the batch means estimator using:

$$\hat{\sigma}_N^2 = \frac{q}{p-1}\sum_{j=0}^{p-1}(\bar{z}^j - \bar{z}_N)^2. \tag{15}$$

The estimator is just the variance of the batch means around the history average $\bar{z}_N$. This statistic has $p-1$ degrees of freedom. As in Jones et al. (2006) and Lang et al. (2019), we take $p = q = \sqrt{N}$ when using this estimator.

**Overlapping batch means variance estimator.**  The *overlapping batch means* (OLBM) estimator Flegal & Jones (2010) has better asymptotic variance than the batch means estimator. The OLBM estimator is conceptually the same, but it uses $N - p + 1$ overlapping batches of size $p$ (rather than disjoint batches) and has $N - p$ degrees of freedom. It can be computed as:

$$\hat{\sigma}_N^2 = \frac{Np}{(N-p)(N-p+1)}\sum_{j=0}^{N-p}(\bar{z}_N - \frac{1}{p}\sum_{i=1}^{p} z_{j+i})^2. \tag{16}$$

## B  THE STATISTICAL TESTS IN SASA AND SASA+

As mentioned in our contributions, there are two main differences between SASA from Lang et al. (2019) and SASA+. First, SASA+ has a much more general "master stationary condition" (10), so it can be applied to any stochastic optimization method with constant hyperparameters (e.g., SGD, stochastic heavy-ball, NAG, QHM, etc.), while the stationary condition in Yaida (2018) and SASA proposed in Lang et al. (2019) (see Equation (11)) only applies to the stochastic heavy-ball method. Second, the statistical tests used in SASA and SASA+ are quite different, as we elaborate below.

The difference starts from a conceptual change from SASA to SASA+. In SASA, one wants to **confidently** detect stationarity. If stationarity is detected, one decreases the learning rate, and otherwise keeps it the same. In SASA+, one wants to **confidently** detect non-stationarity. If non-stationarity is detected, one keeps the learning rate the same, and otherwise decreases. This conceptual change leads to a simpler and more rigorous statistical test in SASA+.

### B.1  EQUIVALENCE TEST IN SASA

To confidently detect stationarity, SASA has to set non-stationarity as null hypothesis and stationarity as the alternative hypothesis. If one confidently rejects the null (non-stationarity), then one can be confident that the process is stationarity. Instead of detecting stationarity, SASA simplifies to only detect the **necessary but not sufficient** stationarity condition (11). Formally, the test in SASA is

$$H_0 : \mathbf{E}[\Delta] \neq 0 \text{ vs. } H_1 : \mathbf{E}[\Delta] = 0, \tag{17}$$

where samples of $\Delta$, i.e., $\Delta_k \triangleq \langle x^k, d^k \rangle - \frac{\alpha}{2}\|d^k\|^2$, are collected along the training process. This kind of test is called an equivalence test in statistics, see, e.g., Streiner (2003). There is no power[1] in the equivalence test (17), i.e., one cannot confidently reject the null hypothesis and prove stationarity

---

[1]In statistical hypothesis testing, power is the ability to reject the null hypothesis when it is false.

at all! Intuitively, even when the process is stationary, with only a finite number of (noisy) samples $\{\Delta_k\}_{k=0}^{N_1}$, the sample mean $\bar{\Delta}_N \neq 0$ (with probability one) is always more likely to be the true mean than the **singleton** 0. In other words, one can not deny that the process is probably infinitely close to stationary but still non-stationary.

To gain power in the equivalence test, one needs to use domain knowledge to define an *equivalence interval*. Formally, the true test in SASA is

$$H_0 : |\mathbf{E}[\Delta]| > \zeta\nu \text{ vs. } H_1 : \mathbf{E}[\Delta] \in [-\zeta\nu, \zeta\nu], \tag{18}$$

where $\zeta\nu$ is the equivalence interval. In English, instead of the usual null hypothesis of not-equal-to-zero in (17), now the null hypothesis is not-equal-to-zero by a margin $\zeta\nu$. In this case, when $\mathbf{E}[\Delta]$'s confidence interval is contained in the equivalence interval, i.e.,

$$\left[\bar{\Delta}_N - t^*_{1-\delta/2}\frac{\hat{\sigma}_N}{\sqrt{N}}, \ \bar{\Delta}_N + t^*_{1-\delta/2}\frac{\hat{\sigma}_N}{\sqrt{N}}\right] \subset [-\zeta\bar{\nu}_N, \zeta\bar{\nu}_N]. \tag{19}$$

we are confident to reject the null hypothesis and to prove/accept $H_1$ (the stationary condition is met within an error tolerance). This is exactly the test in SASA Lang et al. (2019), see its Equation (10). However, this equivalence test requires estimation of $\bar{\nu}_N$ (that estimates the magnitude of $\Delta$) and an additional hyperparameter $\zeta$ that controls the equivalence interval width. In SASA's notation, the equivalence width $\zeta$ is denoted by $\delta$.

Moreover, SASA makes the unjustified assumption that under their null hypothesis ($H_0 : |\mathbf{E}[\Delta]| > \zeta\nu$), the Markov central limit theorem holds. Under their $H_0$, the process is non-stationary, while the construction of $\mathbf{E}[\Delta]$'s confidence interval in Eqn. (19) relies on the (asymptotic) stationarity of the Markov process. Therefore, despite the empirical success of SASA, its statistical test is intrinsically flawed because of this testing setup.

## B.2 Standard test in SASA+

In SASA+, we want to **confidently** detect non-stationarity. If non-stationarity is detected, one keeps the learning rate the same, and otherwise one decreases it. This conceptual change naturally removes the complication of the equivalence test and the intrinsic flaw in SASA.

To confidently detect non-stationarity, SASA+ sets stationarity as the null hypothesis and non-stationarity as the alternative:

$$H_0 : \text{The process is staionary vs. } H_1 : \text{The process is not staionary.} \tag{20}$$

The master stationary condition $\mathbf{E}[\Delta] = 0$ is a necessary condition for stationary of the process, and thus confidently rejecting $\mathbf{E}[\Delta] = 0$ is sufficient to confidently reject the null (stationarity) hypothesis. Moreover, under this null hypothesis, the Central Limit Theorem of Markov Processes exactly holds true (because the process is stationary), validating the construction of $\mathbf{E}[\Delta]$'s confidence interval. Therefore, the intrinsic flaw in SASA is naturally solved in SASA+.

Now we describe the test in SASA+. When the confidence interval of $\mathbf{E}[\Delta]$ does not contain 0, i.e.,

$$0 \notin \left[\bar{\Delta}_N - t^*_{1-\delta/2}\frac{\hat{\sigma}_N}{\sqrt{N}}, \ \bar{\Delta}_N + t^*_{1-\delta/2}\frac{\hat{\sigma}_N}{\sqrt{N}}\right], \tag{21}$$

then we reject the null (stationary) hypothesis and keep the learning rate the same. Otherwise, we decrease the learning rate. Notice that there is no additional hyperparameter $\zeta$.

## B.3 The difference in practice

Although SASA and SASA+ are conceptually quite different, they both use the same confidence interval (see (19) and (21)) for $\mathbf{E}[\Delta]$. In practice, their difference is illustrated in Figure 7. Their difference lies in Case (2) and (3). In Case (3), the process has not reached stationarity yet with high probability, according to the confidence interval of $\mathbf{E}[\Delta]$ (Red). SASA+'s test is confident to reject its (stationary) null hypothesis, so it keeps the learning rate the same. However, SASA decreases its learning rate because it is confident that the stationary condition holds true within its error tolerance (equivalence interval). In this case, SASA makes an error due to its relatively large equivalence

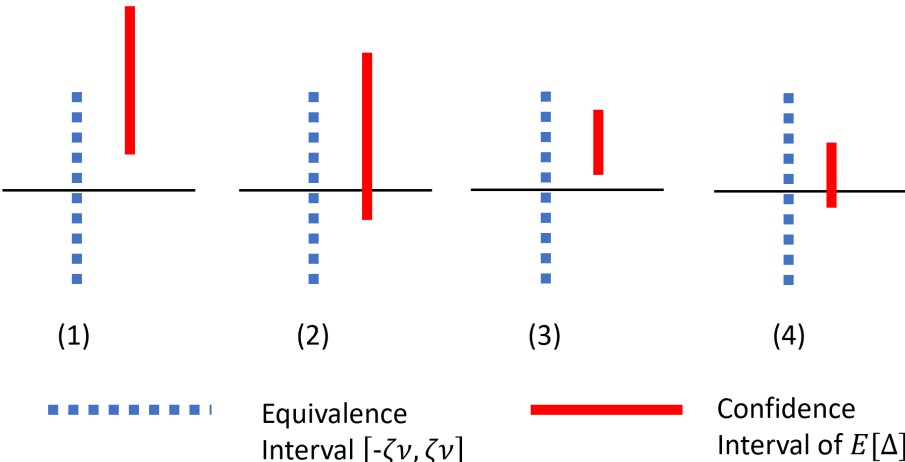

Figure 7: Statistical tests in SASA and SASA+. Case (1): both SASA and SASA+ keep the learning rate. Case (4): both SASA and SASA+ decrease the learning rate. Case (2): SASA keeps the learning rate while SASA+ decreases. Case (3): SASA decreases the learning rate while SASA+ keeps.

interval. In Case (2), SASA+'s test is not confident to reject its (stationary) null hypothesis and so it decreases its learning rate. On the contrary, SASA's test is not confident to claim that the stationary condition holds true within its error tolerance and thus keeps its learning rate. In this sense, SASA+ is more aggressive than SASA in decreasing learning rate.

In numerical experiments on the CIFAR10 and ImageNet datasets, the (width of) the equivalence interval $\zeta\nu$ is typically much smaller than the (width of) the confidence interval of $\mathbf{E}[\Delta]$, see Figure 5(a), 7(a) and 9(a) in Lang et al. (2019). Notice that in those figures, the yellow curve is $\nu$ instead of $\zeta\nu$, and thus the equivalence interval is even smaller. Therefore, Case (3) happens very rarely in those experiments, and this explains the reason why SASA does not make obvious mistakes in decreasing the learning rate. In practice, Case (2) sometimes happens, and thus we can see that SASA+ seems to be slightly more aggressive at decreasing the learning rate than SASA. For example in Figure 1, the black curve (SASA+) is faster at decreasing the learning rate than the green curve (SASA).

