# OpenReview forum: "Statistical Adaptive Stochastic Optimization"
_ICLR.cc/2020/Conference — Reject_

### Official Review · AnonReviewer2 · 2019-10-11
**Official Blind Review #2**

**Rating:** 8

**Review:**

This paper proposes a new way of automatically scheduling the learning rate in stochastic optimization algorithms: Stochastic Approximation with Line-search and Statistical Adaptation (SALSA).
By first introducing a necessary condition for stationarity, the authors use this condition to make a simple statistical test for non-stationarity. Using this test, the authors propose the following strategy for the learning rate schedule in stochastic optimization problems:
(1) They first apply a new Line-Search algorithm (Smoothed Stochastic Line-Search - SSLS) to increase a small initial learning rate until the process becomes stationary according to their statistical test. At this stage, the learning rate is assumed to be optimally initialized for the objective function considered.
(2) The second step is to decrease the learning rate gradually every time the process is stationary again. For this, the authors derived their own version of a Statistical Adaptive Stochastic Approximation algorithm called SASA+ based on their statistical test.
The resulting strategy benefits from being simpler than previous statistical tests while being equally effective empirically.
The authors empirically demonstrated that their new learning rate scheduling mechanism achieves comparable, if not better, accuracy on two image classification tasks with ResNet-18 neural networks. It is important to note that the compared baselines got their parameters slightly fine-tuned, while (according to the authors) the proposed approach was not fine-tuned, and only the default parameters were used. This shows the robustness of the proposed approach against its various parameter settings.

I would accept this submission because the authors propose a new learning rate scheduling mechanism that seems to perform well empirically while being robust against its different initial parameter settings (including the choice of the initial learning rate).

This paper has several novelties: first, it proposes a simpler, yet as effective as previous approaches, Statistical Adaptive Stochastic Approximation (SASA) algorithm based on a new statistical test for non-stationarity. Second, it manages to relax the dependence of SASA algorithms on their optimal initial learning rate by introducing a Smoothed Stochastic Line Search (SSLS) algorithm that is responsible for finding such an optimal initial learning rate. Combined together, these two sub-routine provide a robust mechanism to schedule the learning rate in stochastic optimization problems.

One improvement I could suggest to better motivate the proposed approach is to experiment it not only on Convolutional-based networks with image classification tasks but also on Recurrent-based networks with text datasets. For instance, keeping the ImageNet experiments, the CIFAR-10 experiments could be replaced by an NLP task. This would show that the proposed approach is robust to different types of deep learning problems.

Overall I found the paper well written, and relatively easy to follow, even for non-theoretical practitioners. A few details listed below could improve even more the quality of this paper:
- At the top of page 2, the last sentence of the top paragraph ("However, these learning rate schedules are insufficient ...") requires a citation.
- In Figure 5: it is a little confusing to have the SASA+(NAG) algorithm in both rows: once in the top row, twice in the bottom row. The difference between the two SASA+(NAG) in the bottom row is well explained, but is there any difference between SASA+(NAG) in the top row and the ones in the bottom row?
- In Figure 5 - bottom row, left graph: if all lines are SASA+ algorithm, the legend should be consistent: either add "sasa+" to all lines or remove it from all lines.
- On page 9, ImageNet paragraph: a small typo: "On the other hand, both both SASA+ and SALSA...".

**Experience Assessment:**

I do not know much about this area.

**Review Assessment: Checking Correctness Of Derivations And Theory:**

I did not assess the derivations or theory.

**Review Assessment: Checking Correctness Of Experiments:**

I carefully checked the experiments.

**Review Assessment: Thoroughness In Paper Reading:**

I read the paper thoroughly.

---

> ### Author Response · Authors · 2019-11-12
> **A more detailed caption for Figure 5**
>
> Thank you for your time and feedback.
>  “In Figure 5, is there any difference between SASA+(NAG) in the top row and the ones in the bottom row?”
> There’s no difference between SASA+(NAG) in both rows. They are the same run. The first row is designed to compare different standard algorithms (sgd with \beta=0.9, Adam, SASA, SASA+ and SALSA). We pick one representative from SASA+ and SALSA (because they can work with different stochastic optimization algorithms) to avoid over-crowded figure. The second row is designed to show that SASA+ can be combined with different stochastic optimization algorithms, like heavy ball, NAG and QHM, etc. Therefore, there is one overlap curve in Row 1 and Row 2.
>
> “In Figure 5 - bottom row, left graph: if all lines are SASA+ algorithm, the legend should be consistent: either add "sasa+" to all lines or remove it from all lines.”
> The first three curves (without SASA+) are stochastic optimization algorithms with hand-tuning learning rate, i.e., decrease every 30 epochs as it is shown in Figure 5 Bottom Right figure. The last three curves are SASA+ combined with these 3 algorithms. SASA+ automatically adapt their learning rate (see Figure 5 Bottom Right figure), achieves comparable and even slightly higher testing accuracy (see Figure 5 Bottom Middle figure).
>
> “One improvement I could suggest to better motivate the proposed approach is to experiment it not only on Convolutional-based networks with image classification tasks but also on Recurrent-based networks with text datasets.”
> It is a great suggestion and we will do experiments on RNN and other models as well, and the initial results looking promising.
>
> “At the top of page 2, the last sentence of the top paragraph ("However, these learning rate schedules are insufficient ...") requires a citation.”
> From Lang et al. (2019): "Many former and current state-of-the art results use the constant-and-cut schedules rather than the polynomial decay schedules during training, such as those in image classification [1], object detection [2], machine translation [3], and speech recognition [4]. Additionally, some recent theoretical evidence indicates that in some (strongly convex) scenarios, the constant-and-cut scheme has better finite-time last-iterate convergence performance than other methods [5]."
>
> In practice, these polynomial decay schedules sometimes work with proper tuning of the hyperparameters a, b and c. Therefore, to be rigorous, we have also changed this to “these learning rate schedules still require significant hyperparameter tuning efforts in modern machine learning practice.”
>
> [1] Yanping Huang, Yonglong Cheng, Dehao Chen, HyoukJoong Lee, Jiquan Ngiam, Quoc V Le, and Zhifeng Chen. Gpipe: Efficient training of giant neural networks using pipeline parallelism. arXiv preprint arXiv:1811.06965, 2018.
> [2] Christian Szegedy, Wei Liu, Yangqing Jia, Pierre Sermanet, Scott Reed, Dragomir Anguelov, Dumitru Erhan, Vincent Vanhoucke, and Andrew Rabinovich. Going deeper with convolutions. In Proceedings of the IEEE conference on computer vision and pattern recognition, pages 1–9, 2015.
> [3] Jonas Gehring, Michael Auli, David Grangier, Denis Yarats, and Yann N Dauphin. Convolutional sequence to sequence learning. In Proceedings of the 34th International Conference on Machine Learning-Volume 70, pages 1243–1252. JMLR. org, 2017.
> [4] Dario Amodei, Sundaram Ananthanarayanan, Rishita Anubhai, Jingliang Bai, Eric Battenberg, Carl Case, Jared Casper, Bryan Catanzaro, Qiang Cheng, Guoliang Chen, et al. Deep speech 2: End-to-end speech recognition in english and mandarin. In International conference on machine learning, pages 173–182, 2016.
> [5] Rong Ge, Sham M Kakade, Rahul Kidambi, and Praneeth Netrapalli. The step decay schedule: A near optimal, geometrically decaying learning rate procedure. arXiv preprint arXiv:1904.12838, 2019.

---

### Official Review · AnonReviewer3 · 2019-10-22
**Official Blind Review #3**

**Rating:** 6

**Review:**

The authors explore how stationarity tests can be leveraged to automatically tune the learning rate during training. Their algorithm also add a robust line search algorithm, to reduce the need to tune the initial learning rate. The paper is clear and the literature review is honest and thorough. However, it is unclear to me if the contribution of the authors is enough, as the method used and its presentation are very close to Lang and al. In particular:

- First bullet point on page 2: Because of its conceptual and analytical simplicity, it greatly simplifies implementation and deployment in software packages. It is unclear why the approach proposed by Lang is more complicated to use
- It is a recurrent theme in the paper that the proposed method is a simple interval test compared to a more complicated equivalence test in Lang et al. It is unclear to me what the authors mean by that, as pages 5 and 6 of Lang clearly details a confidence interval test too.
- If SASA+ is indeed just a new presentation of the algorithm detailed by Lang, the line search contribution does not justify a paper in my opinion
- In page 5, "Another major difference is that they set non-stationarity as the null hypothesis and stationarity as the alternative hypothesis (opposite to ours)." I am not sure how the authors arrived to the conclusion that non-stationarity was the null hypothesis in Lang and would appreciate some clarifications on this point. The test used in SASA is a simple t test with a variance corrected to account for the auto-correlation of the gradients.

About the empirical work:

- The experiments seems plausible. Hyper parameters were search for fairly for the competing methods. Adam could have benefited from a finer learning rate schedule, as it is only decreased once compared to several time for the SGD. I would indeed expect a performance gap between Adam and SGD, but I think most of it in this case comes from the one step schedule.
- Line search is performed but no metrics were shown to discuss the computational overhead of evaluating the model and its gradients for different parameters during the search.

SALSA appears to be an already existing algorithm on which a line search was plugged in. The line search part, which appears to be the only contribution, is not discussed enough in my opinion (in terms of computation cost for instance)

To conclude, I think the presented work is too close to the existing literature and that the progress made is very incremental.

EDIT after rebuttal: My concerns have been addressed, I revise my rating from weak reject to weak accept.

**Experience Assessment:**

I have read many papers in this area.

**Review Assessment: Checking Correctness Of Derivations And Theory:**

I assessed the sensibility of the derivations and theory.

**Review Assessment: Checking Correctness Of Experiments:**

I assessed the sensibility of the experiments.

**Review Assessment: Thoroughness In Paper Reading:**

I read the paper thoroughly.

---

> ### Author Response · Authors · 2019-11-12
> **SASA+ is more elegant conceptually and much more generally applicable in practice, than SASA in Lang et al.**
>
> We thank Reviewer #3 for the careful review. It appears that the main concern is the difference between SASA in Lang et al. (2019) and SASA+ proposed in this paper. Due to the page limit of the paper, we cannot elaborate much on the differences, but we make them clear below and in Appendix B of the revised version.
>
> The first difference is that we derived a “master condition for stationary” Equation (10) that applies to any stochastic optimization methods with constant hyperparameters (e.g., SGD, stochastic heavy-ball, NAG, QHM, etc), while the stationary condition in Yaida (2018) and SASA (Lang et al., 2019) (see Equation (11) in our paper) only applies to the stochastic heavy ball method. Moreover, our condition (10) is analytically simpler, because it only requires collecting $x^k$ and $d^k$, whereas SASA’s condition requires $x^k$, $d^k$, and $g^k$. Our condition can be implemented in software packages without knowing how the direction $d^k$ is generated and does not need any specific hyperparameters such as $\beta$ for momentum or $\nu$ for QHM. Therefore, due to the general applicability and simplicity of our condition, “it greatly simplifies implementation and deployment in software packages.” We believe such a generality is a major progress over SASA in Lang et al. (2019) and Yaida (2018).
>
> The second difference is in the statistical tests used by SASA and SASA+. In particular, the statistical test in SASA+ is as below:
> H0: stationary	vs	H1: non-stationary.
> Under the null hypothesis H0, SASA+ computes the likelihood of the stationary condition Equation (10). If the likelihood is low, then we are confident to reject the null hypothesis (stationarity) and do *not* decrease the learning rate. In other words, when it is *not* confident enough to reject the null hypothesis (stationarity), SASA+ decreases the learning rate.
>
> SASA in Lang et al. did not explicitly state its null and alternative hypothesis. According to Equation (10) in Lang et al., its statistical test can be formulated as below:
> H0: $|z| \ge \delta |v|$	vs	H1: $|z| < \delta |v|$.
> SASA computes the likelihood of $|z| \ge \delta |v|$. If the likelihood is low (i.e., its Equation (10)), then SASA is confident to reject the null hypothesis and decrease the learning rate. One can see that $|z| \ge \delta |v|$ is a relaxed version of $|z| > 0$, i.e., the process is non-stationary. Therefore, we claim that SASA sets non-stationarity as the null hypothesis. Moreover, due to its non-stationary null hypothesis, SASA has to introduce the extra hyperparameter $\delta$, which is not needed in the SASA+’s simpler test.
>
> SASA’s test is an “equivalence test” (see SASA’s reference Streiner 2003), whereas the test in SASA+, as you point out, is a simple confidence interval test.
>
> **We put a more detailed discussion in Appendix B of the revised version.**
>
> Combining the above two differences in stationary conditions and statistical tests, SASA+ is more elegant conceptually and much more generally applicable in practice than SASA.
>
> Replies to your other comments:
> “Adam could have benefited from a finer learning rate schedule, as it is only decreased once compared to several time for the SGD. I would indeed expect a performance gap between Adam and SGD, but I think most of it in this case comes from the one step schedule.”
> Adam is designed as an “self-adaptive” stochastic optimization algorithm, like our SASA+ and SALSA. It adapts the learning rate per parameter at every step. In this sense, Adam has the finest learning rate schedule. The fair comparison between Adam and SASA+ is starting both algorithms and letting them self-adapt the learning rate. In fact, we even tune the warm-up for Adam so that it can reach reasonable accuracy for ImageNet, while our SASA+ and SALSA do not need this warm-up at all.
>
>  “Line search is performed but no metrics were shown to discuss the computational overhead of evaluating the model and its gradients for different parameters during the search.”
> When we fix the number of training epochs, the wall-clock time of Smoothed Stochastic Line Search (SSLS) is about 1.5 times of that of the plain stochastic optimization algorithm. This 0.5 computational overhead is due to 2 function evaluations in each line search step on average. Notice that the function evaluation is on the same minibatch and line search only needs the function value without gradient. In SALSA, we switch from SSLS to SALSA at one-third of the total training epochs. Therefore, the total computational overhead caused by using SSLS is only 0.15 of the original total training time.

---

> > ### Comment · AnonReviewer3 · 2019-11-14
> > **Response to the Authors**
> >
> > I thank the Authors for the clarifications.
> > Regarding the null hypothesis comment, it was indeed a mistake from my end.  The new appendix added by the authors clarifies the difference with the previous work of Lang et al.
> > The fairness argument regarding the comparison with Adam is reasonable.
> > Regarding the line search, the "2 function evaluations in each line search step on average" is an important piece of information. I don't know if it is included in the current paper, if not I would suggest to add it. The whole paragraph is actually quite informative and would be a nice addition to the appendix.
> >
> > The authors have provided a reasonable rebuttal to my comments. After further study of the previous literature, I now tend to think that the novelty of this paper is sufficient. As a result, I revise my rating to weak accept.

---

### Official Review · AnonReviewer1 · 2019-10-23
**Official Blind Review #1**

**Rating:** 3

**Review:**

This paper proposes a heuristic method for selecting learning rate schedules for momentum-type methods and evaluates the proposed method on two image classification benchmarks including CIFAR10 and ImageNet. Statistical tests are presented to check the stationarity of the gradient updates. And stochastic line-search methods are proposed to warm up the optimization process during early phases.

Pros:

The main idea is to check the non-stationarity of the iterates and decrease the learning rate if stationarity is detected. To check stationarity, a quadratic approximation for the objective is used. Another procedure for how to decrease the learning rate is proposed using a stochastic line search. The idea makes sense to me.

Cons:

-- The quadratic approximation simply assumes that the Hessian matrix of the objective is identity (Equation 10 in Section 2). I appreciate the simplicity of this formulation. Indeed quadratic forms are good local approximations for any general function. On the other hand, it is quite possible that the Hessian matrix has a low-rank structure or has a sharply decaying spectrum. This kind of scenario naturally arise in high-dimensional settings where the model has lots of parameters. Therefore, it seems to me that further justification (either empirical or theoretical) could help clarify the intuition better.

-- The experimental results compare the proposed method to other well-known methods such as SGD and ADAM. From Figure 1 and Figure 5, it seems that the proposed method performs comparatively to both SGD and ADAM. In particular, it is not obvious from the experimental results that there is a huge benefit obtained from the proposed methods. Hence the experimental results seem a bit incremental to me, as far as I can tell.

More comments:
-- Notation: using $\xi$ to denote the data points seems a bit unconventional.
-- Typos: "Pflug also a devised" -> "Pflug also devised".
-- The current version is significantly over length (by more than 1 page).

**Experience Assessment:**

I have read many papers in this area.

**Review Assessment: Checking Correctness Of Derivations And Theory:**

I assessed the sensibility of the derivations and theory.

**Review Assessment: Checking Correctness Of Experiments:**

I assessed the sensibility of the experiments.

**Review Assessment: Thoroughness In Paper Reading:**

I made a quick assessment of this paper.

---

> ### Author Response · Authors · 2019-11-12
> **Response to reviewer#1's comments.**
>
> We thank Reviewer #1 for the feedback. However, the review comments show major misunderstandings of the method and the goal of the paper, which we elaborate below.
>
> First, we do *not* make any quadratic approximation of the objective function F(x). On the contrary, the advantage of our method is that it applies to all objective functions in the general stochastic optimization form given in Equation (1), regardless of quadraticity or convexity of F(x). We state clearly in the paragraph after Equation (10) on page 3: "This condition holds *exactly* for any stochastic optimization method of the form (6) if it reaches stationarity ... Beyond stationarity, it requires no specific assumption on the loss function or noise model for the stochastic gradients." In fact, a necessary and sufficient condition for stationarity is that Equation (8) holds for any test function $\phi$. Here we derive a necessary condition by setting the test function to be a simple quadratic function $\phi(x) = (1/2)\|x\|^2$ . If we reject any necessary condition, we reject the null hypothesis of stationarity. We could also use any other function as the testing function, including the objective function F(x) itself, but this can result in conditions that are very hard to check without approximation. The test function $\phi$ and the objective function F(x) do not need to be related. Using a quadratic test function leads to the simple necessary condition (10) that works exactly for any objective function F(x).
>
> In addition, our master condition (10) also works for general stochastic optimization algorithm of the form (6) where the direction $d^k$ can be the stochastic gradient, or combined with momentum, or generated by the QHM dynamics in (7), or even more general. And the master condition (10) is still *exact* for testing (non)-stationarity. There is no approximation involved here either.
>
> Second, as we stated in the last part of the introduction, our proposed algorithm SALSA is a highly *autonomous* algorithm to increase/decrease its learning rate automatically, and it matches best performance of *hand-tuned methods* (SGD and Adam) on the two benchmark tasks in this paper. The advantage of SALSA is its automation, not necessarily achieving higher testing accuracy. In fact, for the CIFAR10 and ImageNet dataset, with fixed network architectures (ResNet18), researchers and engineers have extensively *hand-tuned* different kinds of algorithms for years to obtain the accuracy reported in this paper, which may be the limit of the ResNet18 architecture. In our experiments, SALSA robustly achieves this accuracy automatically without tuning hyperparameters. We believe such progress is beyond "incremental."
>
> In addition, the line search procedure we propose (SSLS) tackles another major challenge of running any stochastic gradient type of method: how does one set the initial learning rate? We have shown empirically that SSLS can start from an arbitrarily small learning rate and gradually reach a stable learning rate that matches the initial learning rate of hand-tuned schedules. Thus by combining SSLS and SASA+, our SALSA algorithm is fully automatic in the sense that it can automatically search for a stable learning rate for the initial phase of training and then decrease it to obtain best performance.
>
> For “More comments”:
> In our paper, $\xi$ is not denoting the data point, but the randomness in the stochastic optimization algorithm, e.g., mini-batch sampling in empirical risk minimization. Using $\xi$ to denote the per-step randomness is a standard notation for stochastic optimization.

---

> > ### Comment · AnonReviewer1 · 2019-11-14
> > **Response to the Authors' clarification and further explanations**
> >
> > Hi, thanks for clarifying the questions I asked. They help me better understand the goal of this work.
> >
> > 1- I should say that I'm holding the evaluation to a higher standard, as the "Guidance for Reviewers" suggested. In the revised version, I see that the authors have shortened the submission by 1/2 page.
> >
> > 2- As someone who is outside this area, it's difficult for me to judge the extent of contribution in this work. As I mentioned in the original review, I find the high-level idea of checking stationarity in the stochastic process and use it to set the learning rate interesting.
> >
> > 3- The comments in my original review are intended for the authors to improve the presentation of their work. But I'm disappointed that the author's response seems to try to dismiss my comments as opposed to directly addressing them in some way to improve the clarity of presentation. I'm explaining my comments a bit more here and apologize for any confusion in the original review.
> >
> >     -- I'm still not convinced that the Hessian matrix above eq. (9) can be replaced with the identity matrix exactly. For example, in [1,2], they find that the spectrum of the Hessian has a spiked shape on ResNet for various settings. At the present level of presentation, it is difficult for me to tell what is the differences between the stated claim and the results of [1,2].
> >         -- If there is a proof for this claim, then I think stating the assumptions needed as well as the proof could help clarify why it should be true.
> >
> >     -- I can see that the proposed method matches the "tuned" results of SGD/ADAM, but could you show a running time comparison between the different methods? Also, the experiments are currently focused on image tasks. Providing some evidence in another domain, e.g. on some benchmark language tasks, could consolidate the experimental claim.
> >
> >     -- Regarding $\xi$, there's a huge body of work on stochastic optimization so I think being more specific here could help (e.g. if there is a classic work where the presented setup/notations are following, then providing the citation could help). More generally, I think it could help clarify if the presentation style in Sec. 2 could be more formal, e.g. stating the assumptions and the proofs precisely to show the claim. I believe that this paper could benefit from some more rounds of editing to help the reader better understand the details (especially for people outside the area).
> >
> > [1] Ghorbani, Behrooz, Shankar Krishnan, and Ying Xiao. An Investigation into Neural Net Optimization via Hessian Eigenvalue Density.
> > [2] Papyan, Vardan. The full spectrum of deep net hessians at scale: Dynamics with sample size.

---

### Author Response · Authors · 2019-11-12
**Changes in the revised version**

We thank all reviewers for their comments. Based on the feedback, we made the following changes to the paper.

1. We added a section in the Appendix (Appendix B) to explain the difference between the statistical tests used in SASA in Lang et al. (2019) and SASA+ (this paper). We make clear there the differences on both the conceptual/theoretical side and on the practical side (in addition to the experiments comparing the two algorithms that were already present in the paper, e.g., figures 1 and 5).

2. We added a paragraph at the end of the Smoothed Stochastic Line Search (SSLS) section (Section 3) to discuss the computational cost of  SSLS.

3. We made other small changes according to the comments and corrected typos in the original draft.

---

### Decision · Program_Chairs · 2019-12-19

**Decision:**

Reject

**Comment:**

The paper proposes an approach to automatically tune the learning rate by using a statistical test that detects the stationarity of the learning dynamics. It also proposes a robust line search algorithm to reduce the need to tune the initial learning rate. The statistical test uses a test function which is taken to be a quadratic function in the paper for simplicity, although any choice of test function is valid. Although the method itself is interesting, the empirical benefits over SGD/ADAM seem to be minor.